# RLNet: Adaptive Fusion of 4D Radar and Lidar for 3D Object Detection

Ruoyu Xu[1] and Zhiyu Xiang[1]

Zhejiang University, 38 Zheda Road, Hangzhou, China
{xuruoyu,xiangzy}@zju.edu.cn

**Abstract.** Lidar based 3D object detection has made great progress in recent years and has become the mainstream configuration for autonomous vehicles. However, Lidar can experience substantial performance degradation in the case of adverse weather or long-distance object detection, due to its short wavelength and the limitation of energy emission. 4D millimeter-wave radar is capable of providing 3D point clouds similar to Lidar, with much more robustness against adverse weather conditions. However, 3D object detection with only 4D radar is less satisfactory due to the high sparsity and flickering nature of the measurements. In this paper, we propose a novel 3D object detection method termed RLNet, which effectively integrates 4D radar and Lidar through adaptive feature fusion. An adaptive radar point speed compensation and a modality dropout training strategy are further introduced to improve the performance. RLNet achieves the state-of-the-art performance in the experiments, outperforming baseline method by 7.35 and 2.76 percent in mAP on the popular VoD and ZJUODset dataset, respectively. The code will be available.

**Keywords:** 4D Radar · Adaptive Fusion · 3D Object Detection

## 1 Introduction

As a popular sensor for autonomous driving, Lidar is well-known for its capability of providing accurate 3D information of surrounding environments. As a result, Many successful Lidar-based 3D object detection methods [1–5] have been proposed and achieve state-of-the-art performance in a variety of public datasets [6–8]. However, due to the short wavelength and energy emission limitations, Lidar can experience substantial performance degradation in the case of adverse weather [9] or long-distance object detection [10].

In recent years, 4D millimeter wave radar has received widespread attention. Due to the penetrative nature of millimeter waves, radar can well handle adverse weathers such as rain, snow, and fog, and achieve a longer detection distance [9]. Unlike the traditional 3D radar, 4D radar is capable of providing 3D point clouds similar to Lidar, providing the possibility for accurate 3D object detection. Some public datasets such as Astyx [11], VoD [12], Tj4dradset [13] and ZJUODset [10] have been published to boost the research. However, the high sparsity and noisy

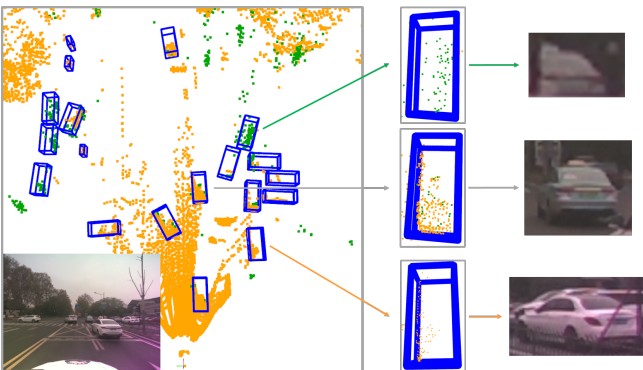

**Fig. 1:** A typical scenario on the ZJUODset [10]. The orange and green points in the left image denote the point clouds acquired by Lidar and 4D radar, respectively. The three enlarged bounding boxes on the right show the exampling objects with only radar points, only Lidar points and both of them, respectively.

nature of the acquired 4D radar point clouds pose a great challenge to the robust detection.

As shown in Fig. 1, the information provided by Lidar and radar sensors can be complementary for the object detection task. Radar can address the limitations of Lidar in the case of part obstruction and objects in the distance, as well as supplying valuable Doppler velocity to facilitate detection of dynamic targets. Current Lidar and radar fusion methods [14, 15] do not well consider the different characteristics of the two modalities during fusion, and they also don't take into account the extreme case of failure of one sensor. Therefore, it is crucial to explore the differences in modalities and design appropriate fusion approaches. To address these problems, we propose a novel 3D object detection method termed RLNet, which effectively fuses 4D radar and Lidar features through adaptive weighting. A radar point speed compensation and a modality dropout training policy are further introduced to improve the detection performance. The experimental results on the VoD dataset [12] and ZJUODset [10] demonstrate the effectiveness of our method.

In summary, our main contributions are as follows:

- We introduce a lightweight Lidar-radar fusion network, which fulfills the 3D object detection task in complex environment by adaptively weighting the importance of 4D radar and Lidar features.
- We propose an effective speed compensation method for radar point cloud preprocessing. We estimate the ego-speed from the raw Doppler speed and then obtain the compensated radial speed for each point.
- We propose a special training method with random modality dropout, which enhances the feature of each single-modality and improves the robustness of the network.

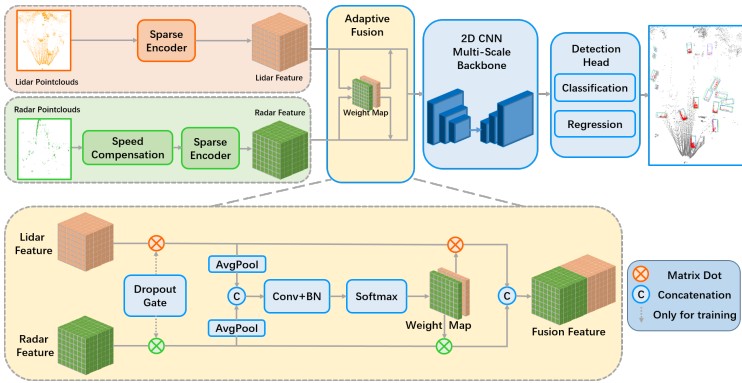

**Fig. 2:** Framework of our RLNet.

– We conduct extensive experiments on the VoD [12] and ZJUODset [10] datasets. Experimental results show that our method achieves state-of-the-art performance.

## 2   Method

In this section, we introduce our RLNet network in detail, which fuses 4D radar and Lidar point clouds for 3D object detection. The network framework is introduced first, followed by description of each module.

### 2.1   Network Framework

We implement our RLNet based on SECOND [2], as shown in Fig. 2. Besides the backbone, the framework consists of a speed compensation module, an adaptive feature fusion module, and a random modality dropout strategy. Before inputting to the network, the 4D radar point clouds are preprocessed by the speed compensation module to obtain the absolute radial velocity for each point. After feature extraction by the sparse encoders, the features from both the radar and Lidar branches are fed into the adaptive feature fusion module, which generates suitable weights to each modal feature before concatenation. We design a random modality dropout strategy during training, thereby enhancing the robustness of the network when the failure of one modality happens. The input for each Lidar point is a 4D-vector with [x,y,z,r], where the first 3 components are the 3D coordinates and the last one is the reflection magnitude. For the 4D radar point, the input is regarded as a 6D-vector, with additional speed and timestamp components over the Lidar point.

### 2.2   Speed compensation

The extra Doppler speed provided by the radar can be valuable for the network to detect and classify the moving objects. However, the raw Doppler speed obtained

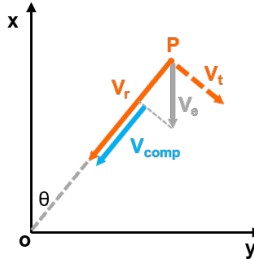

**Fig. 3:** Illustration of speed compensation.

from 4D radar is the relative radial speed to the ego-vehicle. The unknown motion of the ego-vehicle can greatly influence the measured Doppler speed, causing difficulty to the network when utilizing the speed information. To address this issue, we propose a speed compensation module for the radar points based on the assumption that most of the radar points are static in the environment. The process is illustrated in Fig. 3 and described as follows.

1. Assuming that the ego-vehicle moves only on x-y plane and heads along the x direction, we project the raw Doppler speed of the points onto the x-y plane, then project the resulting $V_r$ to x-axis to obtain the relative speed $V_x$ along the vehicle's motion direction with $V_x = V_r/cos\theta$;
2. Assuming the majority of points in the scene are from the background, which is static relative to the world coordinate system, we divide the speed space of $V_x$ into several bins and count the votes in each bin, and the speed value with the highest count is regarded as the estimated ego-vehicle velocity $V_e$;
3. Compensate the radial velocity of each point based on the estimated ego-vehicle velocity, with $V_{comp} = V_r - V_e * cos\theta$. The resulting $V_{comp}$ can better model the speed field of the scene in that most of the static background objects' speeds are zero.

### 2.3   Adaptive Fusion Module

Following the sparse convolution layer on the backbone, the network obtains the Lidar and radar features with each of them having dimensions of B×C×H×W, where B, C, H and W are the batch size, the channel number, the height and width of the feature map, respectively. The simplest way to fuse these two features is concatenation, which combines features along the channel dimension to form a feature of size B×2C×H×W. However, this simple way does not consider the large difference of the sensor characteristics, such as the multi-path noise of the radar data and occlusion sensitiveness of the Lidar data. Simple concatenation of their features can confuse the learning of the network and degrade the performance.

   To tackle this problem, we propose an Adaptive Fusion module to combine the Lidar and radar features by adaptive weights. Specifically, Lidar feature

map $F_L$ and radar feature map $F_R$ are first processed through a channel average pooling layer, yielding two channel feature maps $F_{LA}$ and $F_{RA}$ as follows:

$$F_{LA} = AvgPool(F_L) \tag{1}$$

$$F_{RA} = AvgPool(F_R) \tag{2}$$

Then these two feature maps are concatenated along the channel dimension and fed into a 3×3 convolution layer, followed by batch normalization and softmax. The result is the weight map of $W_L$ and $W_R$, with each having the size of B×H×W:

$$F_{mix} = Concat[F_{LA}, F_{RA}] \tag{3}$$

$$[W_L, W_R] = Softmax(BN(Conv(F_{mix}))) \tag{4}$$

The weight maps represent the importance of each spatial location of the feature maps. The modality features are then weighted and concatenated by Eq.(5), yielding the fused feature map $F_{fusion}$ with size B×2C×H×W:

$$F_{fusion} = Concat[W_L * F_L, W_R * F_R] \tag{5}$$

This fused feature map is then fed into the subsequent backbone to perform 3D object detection.

## 2.4 Random Modality Dropout

The data from multimodal sensors can effectively improve the performance of 3D object detection. However, in some adverse scenarios when a certain sensor is constrained or degraded, the network may receive input from only one sensor. In such case, the full multimodal trained network can experience significant degradation in performance. Inspired by CramNet [16], we introduce random modality dropout strategy during training. The process is described with Eq.(6) and (7), where $F_L'$ and $F_R'$ represent the modal features after dropout gate, $\mathbb{1}(\cdot)$ is the indicator function with outputting values 0 or 1, other parameters are explained in the following.

$$F_L' = \mathbb{1}(p_1 > P_{drop} \parallel p_2 > P_L)F_L \tag{6}$$

$$F_R' = \mathbb{1}(p_1 > P_{drop} \parallel p_2 \leq P_L)F_R \tag{7}$$

First a random keeping probability $p_1$ is generated. If $p_1$ is greater than the dropping probability threshold $P_{drop}$, both features of the two modalities are retained and regular training process proceeds. Otherwise, one of the modality feature should be dropped. In this paper, $P_{drop}$ is set to 0.2.

When deciding which modality feature to drop, another random keeping probability $p_2$ is generated. If $p_2$ is greater than the Lidar's keeping probability threshold $P_L$, the Lidar features are kept and the 4D radar features are dropped; otherwise the Lidar features are dropped in this training epoch. Unlike CramNet [16] which does not consider the difference of the modalities and uses the same

dropout probability for the camera and radar features, we set $P_L$ to 0.2 in this work considering that Lidar features play a dominant role in the fusion. In addition, the experimental results in Section IV suggest that modality dropout strategy can not only mitigate degradation caused by the sensor failure, but also enhance the feature of each modality, thereby promoting the performance of multimodal 3D object detection.

## 3    Experiments

### 3.1    Dataset and Evaluation Metrics

We conduct experiments on the popular VoD dataset and ZJUODset.

**VoD Dataset [12].** The VoD dataset is collected in Delft, which includes a substantial number of car, pedestrian, and cyclist objects to detect. Most of the cars are aligned on the roadside with different extent of occlusions, which pose challenges to the Lidar. We follow the official partition and divide the dataset into training and validation set with 5139 and 1296 frames respectively. Based on the target's distance to the ego-vehicle, VoD separately evaluates the performance on the entire annotated area and the driving corridor. The latter refers to the narrow area within the range of [-4m, 4m] on the x-axis and [0, 25m] on the z-axis (front) in the camera coordinate.

**ZJUODset [10].** ZJUODset collects data on the real traffic scenes of Hangzhou city, aiming at addressing complex and long distance detection requirements for autonomous driving. It collects point clouds acquired from a solid state Livox Lidar and an Oculii Eagle 4D radar, and evaluates the detection performance up to 150 meters. We define the evaluation area as extreme level. 'Easy', 'Moderate' and 'Hard' levels represent objects within 30, 50 and 80 meters, respectively. Within the 3800 annotated frames, we split the first 2660 frames as the training set and the last 1140 frames as the validation set.

AP40 metric is employed for the evaluation. On the VoD Dataset, we use IOU thresholds of 0.5/0.25/0.25 for car, cyclist and pedestrian, respectively, in order to be in line with the previous works. On the ZJUODset, we instead use IOU thresholds of 0.7/0.5/0.5 for car, cyclist and pedestrian respectively, to evaluate the performance with a higher standard than VoD.

### 3.2    Implementation Details

On the VoD Dataset, the entire detection range is set as (0, 51.2m) on the x-axis, (-25.6m, 25.6m) on the y-axis, and (-3m, 2m) on the z-axis in the Lidar coordinate. We set the voxel size to (0.05m, 0.05m, 0.1m) and the maximum number of points in each voxel to 5. On the ZJUODset, we define the detection range as (0, 158.4m) on the x-axis, (-39.6m, 39.6m) on the y-axis, and (-5m, 3m) on the z-axis. The space is partitioned into voxels of (0.075m, 0.075m, 0.2m) for encoding. We use the random flipping along the x-axis and random global scaling with the scaling factor in [0.95,1.05] for data augmentation.

**Table 1:** 3D Detection Results on the VoD dataset. †means we change it to multi-modal method by feature cascade.

| Modality | Method | Entire annotated area AP40@0.5/0.25 | | | | In driving corridor AP40@0.5/0.25 | | | |
|---|---|---|---|---|---|---|---|---|---|
| | | Car | Pedestrian | Cyclist | mAP | Car | Pedestrian | Cyclist | mAP |
| Radar | PointPillars [3] | 34.88 | 31.62 | 63.23 | 43.24 | 72.04 | 41.38 | 88.64 | 67.35 |
| | Second [2] | 35.05 | 29.19 | 55.24 | 39.83 | 73.57 | 43.08 | 83.47 | 66.71 |
| Lidar | PointPillars [3] | 59.11 | 37.71 | 64.49 | 53.77 | 92.35 | 48.02 | 89.08 | 76.48 |
| | Second [2] | 66.95 | 59.90 | 76.88 | 67.91 | 94.69 | 71.15 | 95.63 | 87.16 |
| Radar+Image | RCFusion [19] | 41.70 | 38.95 | 68.31 | 49.65 | 71.87 | 47.50 | 88.33 | 69.23 |
| | LXL [20] | 42.33 | 49.48 | 77.12 | 56.31 | 72.18 | 58.30 | 88.31 | 72.93 |
| Lidar+Radar | PointPillars† [3] | 60.65 | 48.89 | 73.07 | 60.87 | 91.96 | 51.84 | 91.37 | 78.39 |
| | Second† [2] | 68.70 | 63.56 | 81.20 | 71.35 | 94.94 | 72.37 | 94.04 | 87.12 |
| | Interfusion [14] | 55.86 | 49.42 | 70.39 | 58.56 | 84.32 | 55.08 | 91.58 | 76.99 |
| | Ours | **74.26** | **68.98** | **82.57** | **75.26** | **97.35** | **78.10** | **95.82** | **90.42** |

**Table 2:** Experimental Results on the ZJUODset. †means we change it to multi-modal method by feature cascade.

| Modality | Method | 3D Extreme AP40@0.7/0.5 | | | | BEV Extreme AP40@0.7/0.5 | | | |
|---|---|---|---|---|---|---|---|---|---|
| | | Car | Pedestrian | Cyclist | mAP | Car | Pedestrian | Cyclist | mAP |
| Lidar | PointPillars [3] | 42.61 | 10.25 | 29.68 | 27.51 | 62.06 | 12.78 | 35.91 | 36.92 |
| | Second [2] | 42.62 | 13.76 | 39.77 | 32.05 | 62.94 | 21.35 | 48.64 | 44.31 |
| Lidar+Radar | PointPillars† [3] | 44.42 | 15.31 | 40.98 | 33.57 | 62.99 | 20.69 | 50.25 | 44.64 |
| | Second† [2] | 43.14 | 15.36 | 39.64 | 32.71 | 63.10 | 21.67 | 50.93 | 45.24 |
| | Interfusion [14] | **45.16** | 13.05 | 41.39 | 33.20 | **65.11** | 18.17 | 51.22 | 44.83 |
| | Ours | 44.81 | **17.51** | **42.11** | **34.81** | 64.93 | **22.49** | **52.79** | **46.74** |

We implement our RLNet based on mmdetection3d [17] and OpenPCDet [18] framework. We employ the Adam optimizer for parameter updates with an initial learning rate 0.001 and a weight decay factor 0.01. The learning rate is updated using a cyclical decay method, with the maximum learning rate being 0.01 and the minimum being $10^{-7}$. Just as SECOND [2], the loss for the model comprises three components: classification loss, detection regression loss, and angular loss. Specifically, we adopt Focal Loss for classification, Smooth L1 Loss for location regression and Cross-Entropy loss for angular regression.

### 3.3 Experimental Results

The experimental results on the VoD dataset are shown in TABLE 1. Besides comparing to other Lidar+radar methods, we also list methods using other modalities, such as pure radar, pure Lidar and radar+image for reference. As shown in the TABLE 1, our method achieves the best performance among all of the methods, with 3.91% and 3.30% improvements on mAP over the second place method (the feature cascade version of SECOND [2] with Lidar+radar), for the entire annotated area and driving corridor, respectively. Comparing to the pure radar version of SECOND [2], our method has about 35% improvement on mAP in the entire area, showing the importance of the Lidar in the task. Our RLNet also performs much better than the pure Lidar method, showing the critical role of the 4D radar in detecting some hard occluded objects. The qualitative results shown in Fig. 4 also validate this statement. Meanwhile, our

**Table 3:** Ablation Studies(SC refers to Speed Compensation, AF refers to Adaptive Fusion, FC refers to Feature Cascade, and RD refers to Feature Random Dropout).

| Method | | | | BEV mAP40@0.7/0.5 | | | | 3D mAP40@0.7/0.5 | | | |
|---|---|---|---|---|---|---|---|---|---|---|---|
| SC | AF | FC | RD | *Easy* | *Moderate* | *Hard* | *Extreme* | *Easy* | *Moderate* | *Hard* | *Extreme* |
| (a) | ✔ | | | 68.17 | 56.45 | 47.31 | 45.04 | 62.55 | 45.92 | 35.32 | 33.02 |
| (b) ✔ | ✔ | | | 70.07 | 56.74 | 47.47 | 45.40 | 63.66 | 46.32 | 35.76 | 33.48 |
| (c) | ✔ | | | 69.81 | 56.31 | 46.68 | 44.63 | 63.82 | 46.33 | 35.42 | 33.17 |
| (d) ✔ | ✔ | | | 70.42 | 57.21 | 48.53 | 45.99 | 64.09 | 47.10 | 35.80 | 33.77 |
| (e) ✔ | ✔ | | ✔ | **70.67** | **57.38** | **48.89** | **46.74** | **65.03** | **47.34** | **37.15** | **34.81** |

method shows superior performance over the radar+image method like RCFusion [19] and LXL [20], which should thank to the accurate geometric information provided by the Lidar over the image.

Similar phenomenon can be observed on the ZJUODset, as shown in TABLE 2. The difference is that the improvements are much harder to achieve than on the VoD dataset, due to more complex environment, much longer detecting distance requirements (3 times longer than VoD) and higher standard of evaluation metrics of the task. Our RLNet still stands 1.61% higher on mAP over the second best method Interfusion [14], validating the effectiveness of our method. The qualitative results shown in Fig. 5 also reveal less false positive and false negative detections of our method over its counterparts.

### 3.4    Ablation Study

We conduct ablation study on the ZJUODset and the results are in TABLE 3.

**Effects of speed compensation.** We employ a speed compensation strategy to rectify the relative speed caused by the ego-motion of the vehicle. Comparing (a) with (b), or (c) with (d) in TABLE 3, we see that introducing speed compensation can add 0.6∼0.7 percent on 3D mAP, which suggests that removing the influence of ego-motion on the Doppler speed is beneficial for 3D object detection.

**Effects of adaptive fusion module.** The simple feature cascade (FC) cannot fully explore the complementary nature of the Lidar and radar data, resulting in inferior performance. Comparison of (a) with (c) or (b) with (d) in TABLE 3 shows that adaptive feature fusion module (AF) can better fuse the features of the two modalities, with 0.4∼0.7 percent improvements on mAP.

**Effects of random modality dropout.** Comparing (d) with (e) in TABLE 3, we see that introducing random modality dropout (RD) effectively improves the performance, with an increase of 3D and BEV mAP by 1.04% and 0.75%, respectively. Finally, the complete configuration of our method (e) obtains 2.1% higher than the baseline (a) on 3D mAP, revealing the effectiveness of our model design.

# 4   CONCLUSIONS

In this paper we introduce an effective method for 3D object detection by fusing 4D radar point clouds to the popular Lidar. Although both sensors have the similar form of 3D point, the high sparsity and range noise contained in the 4D radar has to be well treated. Based on the popular SECOND [2] backbone, we design speed compensation module for radar points to provide the rectified speed to the network, and propose adaptive fusion module to well balance and enhance the multi-modal features. A special random modality dropout training strategy is further employed to strengthen the robustness of the feature. The experimental results on VoD and ZJUODset datasets demonstrate our success.

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

# A  Related work

## A.1  Lidar-based 3D object detection

The development of 3D object detection based on Lidar has become quite mature to date. VoxelNet [1] divides 3D space into regular voxels for encoding and utilizes 3D convolution to accomplish 3D object detection. However, the employment of 3D convolution introduces substantial computational overhead, making it hard to meet real-time requirements. SECOND [2] proposes sparse 3D convolution, which significantly improves the network's inference speed. PointPillars [3] introduces the concept of pillars and utilizes 2D sparse convolution to further accelerate the network. CVFNet [21] projects point clouds onto the cylindrical view and fuses the original 3D point-level features to accomplish multi-view 3D object detection.

## A.2  Lidar-Camera Fusion for 3D object detection

Introducing image information to Lidar can effectively improve the object detection performance. MV3D [22] uses different views from Lidar point clouds and RGB images as inputs. Candidate boxes are generated from various views, and object detection results are produced by integrated features across the two sensors. PointPainting [23] projects image semantic segmentation results onto 3D point clouds and achieves data fusion at the point level. Very recently, LogoNet [24] fuses features from point clouds and images globally and locally through the attention mechanism.

## A.3  3D Object detection by fusion of traditional 3D Radar

Due to the lack of height information, 3D radar is seldom used alone in the task of 3D object detection. Most of the works focus on the fusion of camera or Lidar to achieve better performance. CenterFusion [25] fuses the features of 3D radar and camera through the Pillar Expansion and Frustum Association. CRAFT [26] designs a soft association method to match the radar points and the image features and proposes a Spatio-Contextual Fusion Transformer to further interact the information between the two sensors. CRN [27] introduces depth estimation and radar occupancy to facilitate view transformation and employs an attention mechanism to conduct feature fusion in BEV. By fusing Lidar, RadarNet [28] employs a voxel-based network to extract features from Lidar and radar point clouds, improving the detection of dynamic objects.

## A.4  3D object detection by fusion of 4D Radar

With similar 3D point clouds to the Lidar, Lidar-based networks can be directly applied to 4D radar without much modification. However, due to the high sparsity and noises with the 4D radar data, work with other modalities is more preferable. RCFusion [19] fuses features of 4D radar and images in the BEV

perspective via attention mechanism. LXL [20] estimates the depth of image through a depth estimation network and then incorporates radar grid occupancy to fuse radar and image features in the BEV perspective. Due to the similarity in the form of point clouds, there has been relatively less prior work focusing on the fusion of 4D radar and Lidar. Interfusion [14] employs an inter-attention mechanism to facilitate information exchange between 4D radar and Lidar at the pillar level, addressing the issue of information loss caused by single sensor. Building on Interfusion, $M^2$-Fusion [29] predicts potential target key points on the feature map, and further refines the pillars around these key points, thereby enhancing the accuracy of 3D object detection. Unlike [14] and [29], we employ voxel level fusion for the two modalities, with an adaptive weighting to balance the fusion. Moreover, we introduce speed compensation for Doppler speed and random modality dropout strategy to enhance the robustness of the network. With minimal additional overhead, our method achieves much higher performance than the baseline method.

## B   More experiments

The model trained by random modality dropout can also better tackle the failure of one modality. To verify this, we test our model by giving only Lidar or radar input, and compare with the model trained by regular process without random dropout. TABLE 4 exhibits the performance gap when only Lidar data is available. As expected, feeding only Lidar modality to the regularly trained RLNet experiences drastic performance drop, while the model trained with random dropout could almost maintain the performance close to the pure Lidar-trained SECOND [2] model. TABLE 5 shows the effect of random modality dropout with radar input only. Considering the long range and the high sparsity of the 4D radar point clouds, only the result of the car category at IOU=0.5 is reported in TABLE 5. Similar phenomenon to TABLE 4 can be observed. However, due to the much lower possibility of feeding radar data only in the RLNet's training process, the gap between RLNet and SECOND becomes larger than that in TABLE 4.

**Table 4:** Effect of Random Modality Dropout with Lidar input only. $\Delta$ represents the performance gap.

| Input | Training | Method | Extreme mAP40@0.7/0.5 | |
| | | | *3D* | *BEV* |
|---|---|---|---|---|
| | Lidar | SECOND [2] | 32.05 | 44.31 |
| Lidar | | RLNET(w/o RD) | 22.55 | 34.44 |
| | Lidar+Radar | RLNET(w/ RD) | 32.25 | 44.15 |
| | | $\Delta$ | +9.70 | +9.71 |

**Table 5:** Effect of Random Modality Dropout with Radar input only. $\Delta$ represents the performance gap.

| Input | Training | Method | Extreme mAP40@0.5/0.25 | |
|---|---|---|---|---|
| | | | **3D** | **BEV** |
| Radar | Radar | SECOND [2] | 27.34 | 37.66 |
| | | RLNET(w/o RD) | 2.91 | 5.88 |
| | Lidar+Radar | RLNET(w/ RD) | 6.82 | 11.48 |
| | | $\Delta$ | **+3.91** | **+5.60** |

# C   Visualization of RLNet

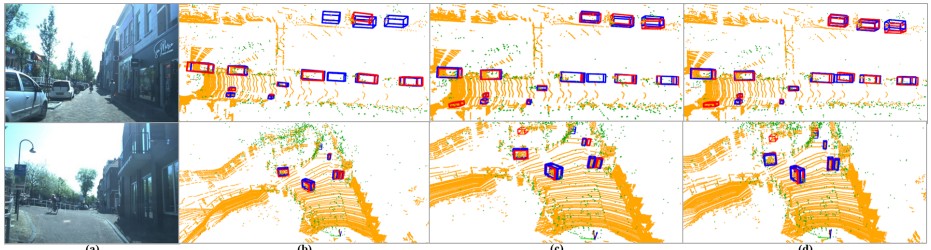

**Fig. 4:** Qualitative results on the VoD dataset. (a) shows the scene images, while (b), (c) and (d) show the corresponding detecting results by the SECOND with only Lidar input, direct cascade of Lidar and radar features, and by our RLNet, respectively. Lidar and radar points are marked orange and green, while the predicted and ground truth bounding boxes are in red and blue, respectively.

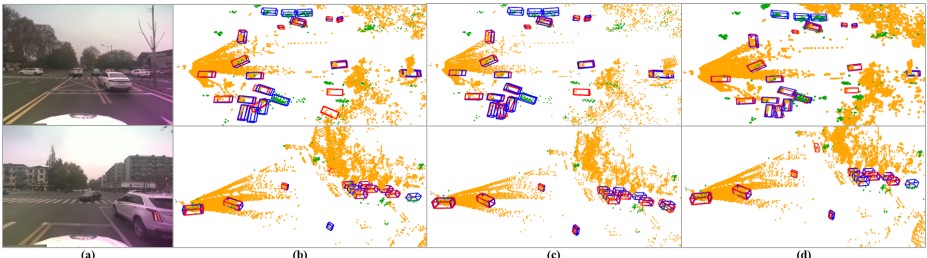

**Fig. 5:** Qualitative results on the ZJUODset dataset. (a) shows the scene images, while (b), (c) and (d) show the corresponding detecting results by the SECOND with only Lidar input, direct cascade of Lidar and radar features, and by our RLNet, respectively. Lidar and radar points are marked orange and green, while the predicted and ground truth bounding boxes are in red and blue, respectively.