# OpenReview forum: "RLNet: Adaptive Fusion of 4D Radar and Lidar for 3D Object Detection"
_thecvf.com/ECCV/2024/Workshop/ROAM — ROAM ECCV 2024 Oral_

### Official Review · Reviewer_StCW · 2024-08-11
**A simple fusion method for lidar and radar sensors**

**Rating:** 7
**Confidence:** 4

**Review:**

This submission presents a novel fusion method of 4D radar and lidar sensor-data. The two modalities have similar 3D representation but are characterized by different point distributions also varying on the environmental conditions. The main message of the submission is clear: the adaptive fusion method proposed by the authors achieves superior results compared to the baseline solutions. The ablation study is clear, it helps evaluating the impact of the ingredients of the final solution.

Feedbacks:
It would be useful to discuss the reasons behind using the two specific datasets for the experiments (VoD, ZJUOD) instead of other publicly available lidar+radar datasets
A more detailed discussion would be helpful about how and why the speed compensation module helps the detection and whether it is related to the synchronization method used in the paper. How the radar timestamps are represented, and how the network is expected to use this information?

---

### Official Review · Reviewer_2xR6 · 2024-08-12
**a simple 4D Radar and Lidar fusion based 3D object detection method**

**Rating:** 6
**Confidence:** 4

**Review:**

This paper presents a simple 4D Radar and Lidar fusion based 3D object detection method. The proposed methods introduced three modules, namely, a speed compensation module, an adaptive feature fusion module, and a random modality dropout module.
The overall framework is very straightforward. So the technical novelty is kind of limited.
The proposed method is validated on the VoD and ZJUODset datasets and the results show some improvements over the baselines.
Why didn't you conduct comparative experiments on the nuScenes dataset which is more commonly used?
Besides, the compared methods are kind of out-dated.

---

### Official Review · Reviewer_k2KG · 2024-08-13
**Review of Submission6**

**Rating:** 6
**Confidence:** 4

**Review:**

In this paper, the authors propose a simple yet effective radar/lidar fusion method for 3D object detection. The core contribution is that a weighting module is proposed to mix the feature of lidar and radar. The experiments are conducted on VoD/ZJUODset datasets. The paper is well-written and easy to follow. The experiments show that the proposed RLNet achieve better performance than single modality input and some other fusion methods.
The main weakness is that the experiments are limited on small, non-mainstream datasets and the compared methods are not SOTA.

---

### Official Review · Reviewer_ZuE1 · 2024-08-17
**Lightweight lidar-radar fusion detection network**

**Rating:** 7
**Confidence:** 3

**Review:**

The paper proposes a lightweight 3D detection network based on lidar and rader input. The proposed method has three novel modules: a speed compensation module for radar data preprocessing, a fusion module to learn weights when fusing lidar and radar features, and a random modality drop training strategy to be more robust to real-world driving data where sensor data quality is compromised.

The paper is well-written and clear. The experiments are conducted on two small datasets and show the proposed method has outperforms lidar-only methods and some other lidar-radar detection works.

I believe that the experiments were not conducted on e.g., nuscenes because the radar data is the traditional 3D radar data that doesn't have height information, and other fusion methods (e.g., CenterFusion) are developed based on the 3D radar instead of the 4D radar that this work focuses on.

---

### Official Review · Reviewer_qwt1 · 2024-08-17
**review of lidar-radar fusion detection**

**Rating:** 6
**Confidence:** 4

**Review:**

The paper proposes a 3D object detection method termed RLNet, which effectively integrates 4D Radar and Lidar through adaptive feature fusion. The framework is relatively simple and has limited innovation.

pros:
1.The paper is well organized.
2.The experimental result outperforms baseline method by 7.35 and 2.76 percent in mAP on the VoD and ZJUODset dataset.

cons:
1.Existing experiments cannot reflect the excellent performance of the algorithm. It is recommended to compare with more updated Lidar+Radar 3D detection methods.
2.The Lidar+Radar version Second implemented by the author has very similar performance with the Lidar version Second, and the baseline seems a bit easy.
It is recommended to analyze the reasons for the poor vehicle detection results on the ZJUODset dataset.

---

### Official Review · Reviewer_kyeL · 2024-08-17
**Review for 'RLNet: Adaptive Fusion of 4D Radar and Lidar for 3D Object Detection'**

**Rating:** 7
**Confidence:** 4

**Review:**

In this paper, the authors present an approach for 3D object detection that fuses 4D radar and LiDAR data using adaptive feature fusion. The paper's motivation stems from the complementary nature of these two sensing modalities: LiDAR provides precise 3D spatial information, while 4D radar offers robustness in adverse weather conditions and over longer distances. However, both sensors have inherent limitations that the authors aim to address this through a new fusion technique.

The adaptive feature fusion module, which dynamically weights the importance of radar and LiDAR features, is a significant contribution. This method effectively handles the distinct characteristics of each modality. Additionally, the inclusion of radar point speed compensation and the random modality dropout during training are practical enhancements that contribute to the robustness of the model. The ablation studies, which test the impact of these individual architectural improvements, are particularly insightful.

While the paper demonstrates strong results on two datasets, it would be beneficial to explore how and under what conditions this method generalizes to other datasets.

---

### Official Review · Reviewer_iQV5 · 2024-08-21
**Review of Submission 6**

**Rating:** 6
**Confidence:** 4

**Review:**

RLNet, the core contributions are an adaptive Fusion module to combine the Lidar and radar features by adaptive weights and a special training method with random modality dropout (Inspired by CramNet when a certain sensor is constrained or degraded). Their experiments achieve better performance than single modality input and some other fusion methods (Lidar, Radar, Radar+Image and Lidar+Radar); their experiments also reveal fewer false positive and false negative detections over its counterparts, however, the results were only displayed in the VoD and ZJUODset datasets, did you use other datasets such as BDD100K or nuScenes? how do you calculate the 0.2 from the dropout probability threshold? (eq. 6 and 7). How are the additional speed and timestamps represented in the 6D-vector input?

---

### Decision · Program_Chairs · 2024-08-22

**Decision:**

Accept (Oral)

**Comment:**

The average score of the paper given all the reviews received before the deadline was higher than 5.5 (1 is lowest, 10 is highest), therefore the paper is accepted. The Authors are encouraged to consider feedback for the camera ready version of the paper due on August 31st.